# *Lactiplantibacillus plantarum* from Unexplored Tunisian Ecological Niches: Antimicrobial Potential, Probiotic and Food Applications

**DOI:** 10.3390/microorganisms11112679

**Published:** 2023-10-31

**Authors:** Hiba Selmi, Maria Teresa Rocchetti, Vittorio Capozzi, Teresa Semedo-Lemsaddek, Daniela Fiocco, Giuseppe Spano, Ferid Abidi

**Affiliations:** 1Faculty of Sciences of Bizerte, University of Carthage, Zarzouna, Bizerte 7021, Tunisia; 2Laboratory of Protein Engineering and Bioactive Molecules (LIP-MB), National Institute of Applied Sciences and Technology, University of Carthage, Carthage 1054, Tunisia; ferid.abidi@insat.ucar.tn; 3Department of Clinical and Experimental Medicine, University of Foggia, 71122 Foggia, Italy; mariateresa.rocchetti@unifg.it (M.T.R.); daniela.fiocco@unifg.it (D.F.); 4Institute of Sciences of Food Production, National Research Council (CNR) of Italy, c/o CS-DAT, Via Michele Protano, 71122 Foggia, Italy; vittorio.capozzi@ispa.cnr.it; 5Centre for Interdisciplinary Research in Animal Health (CIISA), Faculty of Veterinary Medicine, University of Lisbon, Av. da Universidade Técnica de Lisboa, 1300-477 Lisbon, Portugal; tlemsaddek@fmv.ulisboa.pt; 6Associate Laboratory for Animal and Veterinary Sciences (AL4AnimalS), 1300-477 Lisbon, Portugal; 7Department of Agriculture Food Natural Science Engineering (DAFNE), University of Foggia, 71122 Foggia, Italy

**Keywords:** lactic acid bacteria, probiotic, biocontrol, niches, Tunisian, tomato

## Abstract

The continued exploration of the diversity of lactic acid bacteria in little-studied ecological niches represents a fundamental activity to understand the diffusion and biotechnological significance of this heterogeneous class of prokaryotes. In this study, *Lactiplantibacillus plantarum* (*Lpb. plantarum*) strains were isolated from Tunisian vegetable sources, including fermented olive and fermented pepper, and from dead locust intestines, which were subsequently evaluated for their antimicrobial activity against foodborne pathogenic bacteria, including *Escherichia coli* O157:H7 CECT 4267 and *Listeria monocytogenes* CECT 4031, as well as against some fungi, including *Penicillium expansum*, *Aspergilus niger*, and *Botrytis cinerea*. In addition, their resistance to oro-gastro-intestinal transit, aggregation capabilities, biofilm production capacity, adhesion to human enterocyte-like cells, and cytotoxicity to colorectal adenocarcinoma cell line were determined. Further, adhesion to tomatoes and the biocontrol potential of this model food matrix were analyzed. It was found that all the strains were able to inhibit the indicator growth, mostly through organic acid production. Furthermore, these strains showed promising probiotic traits, including in vitro tolerance to oro-gastrointestinal conditions, and adhesion to abiotic surfaces and Caco-2 cells. Moreover, all tested *Lpb. plantarum* strains were able to adhere to tomatoes with similar rates (4.0–6.0 LogCFU/g tomato). The co-culture of LAB strains with pathogens on tomatoes showed that *Lpb. plantarum* could be a good candidate to control pathogen growth. Nonetheless, further studies are needed to guarantee their use as probiotic strains for biocontrol on food matrices.

## 1. Introduction

Lactic acid bacteria (LAB) are a group of Gram-positive, non-spore-forming, cocci or rod, catalase-negative microorganisms [1] widely used in fermentation, probiotics, and the food/beverage manufacturing industries. LAB are generally regarded as safe (GRAS) by the United States Food and Drug Administration (FDA) and boast the qualified presumption of safety (QPS), as assessed by the European Food Safety Authority (EFSA) [2]. Moreover, LAB have been widely used to improve the taste, texture, and nutritional properties of a wide variety of foods, including vegetables, meat, dairy, and cereal substrates [3], and can extend their shelf life by producing organic acids, carbon dioxide, and antimicrobial peptides [4].

In the food industry, LAB have become increasingly important, and many studies have focused on the possible use of bacteriocin-producing species as alternatives to chemical preservatives in foods [5]. Indeed, certain LAB strains exhibit antimicrobial activity against foodborne pathogens, including bacteria, yeast, and filamentous fungi. Furthermore, in recent years, many authors proved that LAB have the ability to neutralize several undesired microbes, including *Clostridium* spp., *Enterococcus faecalis*, and *Listeria monocytogenes* [6].

Thanks to their fermentation ability, LAB are used in the manufacturing of many foods, such as dairy products, sausages, cucumber pickles, and olives [7]. Indeed, LAB can be divided into starter cultures, used to drive biochemical changes in primary relevance for food fermentation, and non-starter cultures, usually deriving from the autochthonous microbiota of the food matrix [8] and contributing in secondary aspects to the organoleptic characteristics of the final product [9]. The antimicrobial potential of non-starter LAB isolated from fermented foods has been substantiated by scientific literature and involves the production of organic acids, hydrogen peroxide, and bacteriocins. 

In addition to antimicrobial and fermentation capacities, some LAB also exhibit probiotic features. According to the Food and Agriculture Organization (FAO) of the United Nations and the World Health Organization (WHO), probiotics are “live microorganisms which, when administered in adequate amounts, confer health benefits on the host” [10]. It has been confirmed that the consumption of probiotics can reduce cholesterol serum levels, prevent diarrhoea through the enhancement of the intestinal barrier, and decrease irritable bowel syndrome symptoms [11]. To be applied as a probiotic and to function appropriately, a microbial strain must exhibit several characteristics, such as gastrointestinal tolerance and the ability to colonize the human host. Furthermore, the probiotic candidate should be safe for humans, e.g., with an antibiotic sensitivity phenotype [12].

*Lactiplantibacillus plantarum* (*Lpb. plantarum*) is a LAB with a long history of protechnological use in the food sector, in food fermentations, and for the design of protective cultures [13,14]. This species can adapt to a variety of niches and is widely distributed in the environment, i.e., it can be found in dairy products, in the gastrointestinal tracts of humans and animals, and in meat, fish, and fermented vegetables [15]. Increasing evidence also corroborates the probiotic properties of *Lpb. plantarum* strains [16,17], thus broadening the range of its applications. To date, most studies underline the safety attributes of *Lpb. plantarum* [18,19], supporting the industrial interest in this species. Recent papers also highlight some *Lpb. plantarum* properties that make it intriguing for biomedical purposes, e.g., its capacity to regulate the enteric microbiota and alleviate liver disease [20] and its anticancer potential [20]. Moreover, some publications hint at the use of *Lpb. plantarum* for the bio-suppression of pathogens in food models [21,22].

Fruits and vegetables are important components of a healthy diet, and their consumption helps prevent a wide range of diseases [23]. Thus, both the WHO and FAO recommend the intake of a specific dose of vegetables and fruits *per* day. The tomato is highly consumed worldwide; it is economically considered the second-most important fruit or crop [24]. In 2020, Tunisia ranked tenth in the world in producing tomatoes. The production of tomatoes was estimated at around 1.4 million metric tons in the last four years by the Statista Research Department [25]. Tomatoes contain a high amount of fiber, oligosaccharides, and polysaccharides, which act as prebiotics in the gut [26]. However, tomatoes can be easily contaminated and spoil very fast due to contaminated irrigation water during transporting and storage, causing economic losses and serious health issues related to foodborne diseases [27]. Consequently, there is a need for intervention technologies and techniques to reduce/prevent tomato contamination [28]. To face food contamination/spoilage, several methodologies have been investigated and applied over the years; however, these methods are not applicable to all foods and can alter the sensory properties of the final product [29,30]. Therefore, bio-antimicrobial agents, such as antagonistic bacteria, have emerged as alternatives recently. In this regard, *Pediococcus pentosaceus* [31]*, Lbp. plantarum*, *Lactococcus lactis* [32], and *Latilactobacillus graminis* [33] have been described as promising species for preventing biological contamination and spoilage of fresh vegetables.

Overall, the aims of the present study were to isolate new LAB strains from unexplored Tunisian sources and evaluate their capability to colonize tomato surfaces and inhibit the adhesion of pathogenic *Escherichia coli* O157:H7 CECT 4267 and *Listeria monocytogenes* CECT 4031 with co-culture on the food matrix. Further, their potential use for the design of probiotic cultures was investigated. 

## 2. Materials and Methods

### 2.1. Isolation of LAB and Preliminary Screening for Antimicrobial Activity

LAB strains used in this study were isolated in Tunisia from a wide variety of niches, including different types of homemade traditional food, fermented olives, fermented peppers, and the intestines of dead locust Solitaria (*Tettigonia viridissima*) (see below (Table 1)). The samples were serially diluted in physiologic solution, plated on De Man–Rogosa–Sharpe (MRS), and incubated at 37 °C. Gram staining, catalase tests, and microscope examination of the cell morphology were used to achieve phenotypic identification. Growth in MRS broth and pH variation were also evaluated. The purified LAB isolates were stored in glycerol 20% (*v*/*v*) at −80 °C.

The LAB isolates were screened for their antibacterial activity against foodborne pathogens, such as *Escherichia coli* O157:H7 CECT 4267 and *Listeria monocytogenes* CECT 4031, using the modified agar well method [34]. Briefly, wells were created in Luria–Bertani medium (LB), and 24 h crude bacterial cultures cultivated in MRS broth of the isolated microbes were poured into each well. After 24 h of incubation at 37 °C, LAB strains were discriminated based on inhibition halos around the wells. Six selected strains were further investigated for their biochemical features and stress response in various conditions (e.g., acidic condition, osmotic stress [35]) and for some probiotic features, as follows. 

*Enterococcus faecalis* CECT 795 [36], *Lactobacillus reuteri* DSM 17938 [37], *Enterococcus faecalis* V583 [38,39], and *Lpb. plantarum* 299 V [40] were used as control strains for the hemolytic assay, oro-gastro-intestinal transit tolerance assay, biofilm formation assay, and the adhesion to human enterocyte-like cells, respectively. *Lpb. plantarum* 299 V and *L. reuteri* DSM 17938 were routinely cultivated in MRS medium at 37 °C, while *E. faecalis* CECT 795 and *E. faecalis* V583 were cultured in Brain Heart Infusion medium (BHI) at 37 °C.

### 2.2. Molecular Identification of Selected Isolates

The six isolates with the best antibacterial performances and from unusual sources were selected for further characterization and identification using sequencing of the 16S rRNA gene. The genomic DNA of the strains was purified using a genomic DNA extraction kit (Mericon, Hilden, Germany), following the manufacturer’s instructions. The genomic DNA was used as a template to amplify and sequence 16S rDNA, resulting in species identification (OR431596-OR431601).

### 2.3. Antimicrobial Activity and Partial Characterization of the Antimicrobial Agent

The selected LAB strains were screened for antagonistic activity using the agar diffusion method (well method) as described above (Section 2.1) using *L. monocytogenes* and *E. coli* as indicators. To determine the antifungal spectrum of the selected LAB isolates, the overlay method was performed, according to Russo et al. [41], using *Penicillium expansum* CECT 2278, *Aspergilus niger* CECT 2805, *Fusarium culmorum* CECT 2148, *Saccharomyces cerevisiae*, and *Botrytis cinerea* CECT 20973 as targets. At the late exponential phase, 5 µL of each LAB culture was spotted on MRS agar plates and incubated at 30 °C for 24 h. Then, the plates were overlaid with 10 mL of malt extract (Oxoid) soft agar (0.75% agar) inoculated (1:100 *v*/*v*) with a suspension containing approximately 1 × 10^6^ spores/mL of each fungal species. 

Regarding the partial characterization of the antibacterial agents: LAB culture, crude cell-free supernatant (CFS), neutralized CFS (pH = 6.0), heat-treated CFS (121 °C 10 min), crude cells, and heated cells were prepared as follows. CFS obtained with filtration (0.24 µm) of 24 h LAB cultures were neutralized with 1 M NaOH to pH = 6.0 (to eliminate the antibacterial effect of organic acids) and subsequently heat-treated (121 °C, 10 min) (to neutralize thermosensitive proteins) [42]. LAB cells obtained after centrifugation (10,000× *g*, 5 min) were washed twice with sterile PBS pH = 7 and resuspended in the same volume of the previous LAB culture in PBS pH = 7; later, the cells were treated with heat (121 °C, 10 min). After 24 h of incubation at 37 °C, the halos of inhibition around the wells were measured. The organic acids of the CFSs were quantified using an HPLC Spectra System P1000 XR (Thermo Electron Corporation, Madison, WI, USA) [43].

### 2.4. Safety Features: Antibiotic Resistance and Hemolytic Activity

The antibiotic susceptibility of lactobacilli strains was determined using a disc diffusion assay according to CLSI guidelines [44]. In brief, antibiotics, i.e., ampicillin (10 μg), oxacillin (1 μg), amikacin (30 μg), gentamicin (10 μg), tetracycline (30 μg), chloramphenicol (30 μg), and clindamycin (2 μg), were placed on the surface of plates and incubated at 37 °C for 24 h. The inhibition zone diameters were measured, and susceptibility was expressed as resistant (R), susceptible (S), and intermediate (I), as mentioned [45].

The hemolytic activity was assessed by streaking fresh culture of LAB on Columbia 5% blood agar plates, after incubation for 24 to 48 h at 37 °C [7]. Strains showing a transparent halo or a green-hued halo were considered hemolytic and classified as β-hemolytic and α-hemolytic, respectively, whereas those presenting no halo were considered non-hemolytic and were classified as γ-hemolytic. *Enterococcus faecalis* CECT 795 was used as a negative control (γ-hemolytic).

### 2.5. Autoaggregation and Co-Aggregation Assays

To determine auto-aggregation, we followed previously described procedures with some modifications [46]. Briefly, 5 mL of bacterial suspension was vortexed for 10 s, and then the suspension was incubated for 4 h at 37 °C. The absorbance of the supernatant after 2 h of incubation was subsequently measured (OD_t_) starting from t0. The auto-aggregation percentage was calculated as
A = [1 − (OD_t_/OD_t0_)] × 100.

For the co-aggregation assay [47], 4 mL of each LAB strain and 4 mL of each pathogen culture (*E. coli* or *L. monocytogenes*) were mixed, vortexed for 10 s, and incubated for 4 h at 37 °C. Each control tube contained 4 mL of each single bacterial suspension (i.e., the LAB strain or the pathogen). The absorbance of each mixed suspension was then measured at 600 nm (OD_mix_) and compared with those of the control tubes containing the LAB strain (OD_strain_) and the specific pathogen (OD_pathogen_) at 2 h of incubation. 

Co-aggregation (%) was calculated as C= [1 − OD_mix_/(OD_strain_ + OD_pathogen_)/2] × 100.

### 2.6. In Vitro Oro-Gastrointestinal Transit Tolerance Assay

The oro-gastrointestinal (OGI) transit tolerance assay was performed according to Gheziel et al. [48], with some modifications (Figure 1). Briefly, LAB strains were grown until the mid-exponential phase (OD_600 nm_ = 1) and then centrifuged (10,000× *g*, 5 min) and resuspended into sterile electrolyte solution (6.2 g/L NaCl, 2.2 g/L KCl, 0.22 g/L CaCl_2_, 1.2 g/L NaHCO_3_). To simulate the saliva condition in vitro, bacterial suspensions were incubated at 37 °C for 5 min, with 150 mg/L lysozyme at pH = 6.0. Then, the culture was centrifuged in order to remove the first solution and to simulate gastric stress by adding the second solution (pH = 3) containing pepsin 3 g/L. Later, the second solution was removed with a second centrifugation (10,000 × *g*, 5 min), the pH was decreased to 2.0 using a third solution, and pepsin 3 g/L was added, followed by incubation for 30 min at 37 °C. Finally, intestinal stress was simulated using a fourth solution (pH = 6.5) containing 3 g/L bile salts and 1 g/L pancreatin. Finally, the samples were diluted (1:1 *v*/*v*) with a sterile electrolyte solution to mimic the large intestine and incubated for 1 h at 37 °C. 

Relative viability was calculated by spreading serial dilutions from bacterial samples on MRS agar at different steps of the simulated transit. The survival rate was calculated by comparing colony-forming units (CFUs) from control and OGI-stressed bacterial samples. The survival rate was expressed as LOG (CFUt(1–5)/CFUt0). The results were further compared to that of *Lactobacillus reuteri* DSM 17938 [49], a probiotic strain with good tolerance toward gastrointestinal stress conditions, which was used as a positive control for this experiment.

### 2.7. Biofilm Formation Assay

The production of biofilms was evaluated on 96-well polystyrene microtiter plates [50]. First, the wells were filled with 200 µL of MRS medium, and the absorbance of bacterial suspensions in Maximum Recovery Diluent (MRD) (Liofilchem) was adjusted to 0.5 MacFarland in order to standardize the number of bacteria (10^7^–10^8^ CFU/mL), and 1% of overnight culture was added to each well. The plates were incubated for 24, 48, and 72 h at 37 °C. To quantify the biofilm formation, the well content was carefully removed, and 100 µL of a 0.1% (*v*/*v*) crystal violet solution was added to each well and held at ambient temperature for 15 min; then, the wells were gently washed three times with MRD. Then, the dye bound to adherent cells was removed with 100 µL of acetone/ethanol mixture (80:20), and absorbance was measured at 595 nm.

In order to detect/estimate the viability of adherent LAB cells in the wells, the redox dye resazurin was used [51]. Briefly, after 3 washes of the 96-well plate cultures with MDR, the plates were dried under a laminar flow chamber for 1 h, and then 100 µL of freshly prepared resazurin solution (8 µg/mL) was added. The optical density (OD) of each well was measured at 595 nm using a microplate reader (SUNRISE, Serial number 1708003498, XFLUOR4 Version V 4.51). *Enterococcus faecalis* V583, growing in Brain Heart Infusion medium (BHI) was used as a positive control for biofilm production [52,53]. Each assay was performed in three replicates and conducted three individual times on different days under the same conditions, and the negative control was performed in uninoculated MRS broth and BHI. The cut-off (ODC) was defined as the mean OD value of the negative control. Based on the OD, strains were classified as non-biofilm producers (OD ≤ ODC) or weak (ODC < OD ≤ 2 × ODC), moderate (2 × ODC < OD ≤ 4 × ODC), or strong biofilm producers (4 × ODC < OD) [54].

### 2.8. Cytotoxic Effects of Lactiplantibacillus Plantarum Strains

All LAB strains were grown in 100 mL MRS broth for 24 h (reaching stationary phase) at 37 °C. In order to obtain CFS (cell-free supernatant), we followed the same protocol indicated in (part 2.3) and then lyophilized the samples. The lyophilized samples were dissolved in sterile distilled water to attain the desired concentrations from 0.5 mg/mL to 10 mg/mL. 

For the cytotoxicity assay, human colorectal adenocarcinoma LS174T (CL-188, ATCC, Manassas, VA, USA) cells were used [55]. Briefly, 50 µL of containing ten thousand cells were seeded per well in 96-well plates containing RPMI (Gibco™, Cergy-Pontoise, France; Sigma, St. Louis, MO, USA) supplemented with 10% fetal bovine serum (FBS), 1% L-glutamine, and 100 IU/mL penicillin/streptomycin and incubated for 24 h for cells to adhere properly. 

After 24 h, 100 μL of fresh medium with different concentrations of lyophilized supernatant was added and incubated for 24 h and for 72 h. The medium was removed, and 50 µL of MTT (0.5 mg/mL) was added to all wells. After 2 h of incubation at 37 °C, 100 µL of dimethylsulfoside (DMSO) was added, and the absorbance was read. The optical density was measured at 560 nm. The assay was repeated at least in three independent experiments in triplicate. The percentage of cell viability was calculated using the following equation: A_sample_ − A_b_/A_control_ × 100, where A_sample_ represents the absorbance of treated cells with CFS, A_b_ represents the absorbance of the cell medium (DMEM) with CFS, and A_control_ represents the absorbance of cells with LAB medium (MRS). The concentration for 50% of growth (IC_50_) was determined.

### 2.9. Adhesion to Human Enterocyte-Like Cells

The adhesion capacity was investigated using human Caco-2 cell lines as previously described [56]. Cells were grown in a controlled atmosphere of 5% CO_2_ at 37 °C in Dulbecco’s Modified Eagle’s Minimal Essential Medium (DMEM) (Gibco. Carlsbad, CA, USA), supplemented with 2 mM L-glutamine, 50 U mL^−1^ penicillin, 50 U mL^−1^ streptomycin, and 10% (*v*/*v*) heat-inactivated fetal bovine serum (FBS). Caco-2 (2 × 10^5^ cells per well) was seeded in 96-well cell culture plates and grown for 2 weeks, changing the medium three times a week to obtain steady monolayers. Twenty-four hours prior to the adhesion assay, the growth medium was replaced with absolute DMEM. From each bacterial strain, 100 uL of mid-exponential phase cultures (OD600 nm = 0.6–0.8, corresponding to 5 × 10^8^ CFU mL^−1^) were centrifuged, washed with sterile PBS, resuspended in the same volume of absolute DMEM, and finally, incubated with Caco-2 cells in a ratio of 1000:1 (bacteria:Caco-2 cells). After 1 h of incubation at 37 °C (5% CO_2_), Caco-2 monolayers from the test wells were washed 3 times with PBS to remove non-adherent bacteria and then detached with incubation at 37 °C for 10 min in the presence of trypsin/EDTA (0.05%, Sigma). Trypsin-treated samples were resuspended in PBS, serially diluted, and plated onto MRS agar to enumerate the adherent bacteria. Bacterial CFU obtained from the washed wells (i.e., comprising Caco2 cell-adhering bacteria only) were compared with those obtained with trypsinization from control unwashed wells (i.e., comprising total bacteria, both adhering and non-adhering ones), and the adhesion percentages were calculated as [(CFU)_washed well_/(CFU)_unwashed well_] × 100. The commercial *Lpb. plantarum* 299 V strain, previously proven to possess excellent adhesion abilities [57,58], was used as a positive control. Adhesion assays were conducted at least in three independent experiments in triplicate.

### 2.10. Application in the Food Model

#### 2.10.1. Adhesion on the Tomato Surface

An adhesion assay was performed on whole grape tomatoes (*Solanum lycopersicum L*.), as represented in the figure below (Figure 2) [59]. Fresh tomatoes were purchased from a local supermarket in Italy the day before the experiment and kept in the refrigerator overnight at 4 °C. On the day of the experiment, pericarp-free tomatoes without visible defects, such as bruises and injuries, were selected, washed with distilled water, placed inside the laminar flow cabinet, and irradiated with UV light for 15 min before being inoculated.

Fresh LAB cultures were washed twice with PBS buffer pH = 7 and were adjusted to a final concentration of 1 × 10^8^ cells/mL by resuspending the pellets in PBS to an OD of 0.5 at 600 nm. Fifty microliters of culture suspension were carefully spotted on the stem scar sites of grape tomatoes at ambient air temperature. The inoculated tomatoes were kept in the laminar flow cabinet for 2 h to allow bacterial attachment to the tomato scar. For the microbial enumeration [59], tomato stem scars were aseptically excised from the fruit, combined with physiological water 0.85% in 1/8 ratio (weight/volume), and carefully mixed using a vortex for 2 min. After that, serial decimal dilutions of the homogenate were prepared, inoculated in MRS plates (in triplicate), and incubated for 24 h at 37 °C. For the control, PBS buffer pH = 7.0 was used for tomato inoculation, as mentioned above.

#### 2.10.2. Biocontrol/Pathogen Antagonism on Tomatoes

An antagonism assay was performed as described above for LAB adhesion to the tomato surface by co-inoculating LAB strains and *Escherichia coli* O157:H7 CECT 4267 or *Listeria monocytogenes* CECT 4031. Each LAB was mixed with the corresponding pathogen and inoculated simultaneously on the tomato pericarp by spotting in sterile conditions. The mix of LAB culture and pathogen was prepared previously, as mentioned in [60]: each culture was adjusted to a final cell concentration of 10^9^–10^8^ /mL and then washed twice by suspending the pellet in PBS to an optical density of 0.5 for LAB and 0.2 for the pathogens at 600 nm. After inoculation, the fruits were kept in a laminar flow cabinet for 2 h to allow bacterial attachment to the tomato stem scar and then stored in sterile bags at room temperature.

For microbial quantification MRS agar, Sorbitol McConkey Agar (SMAC) and Polymyxin Acriflavine Aithium Chloride ceftazidime Aesculin Mannitol agar (PALCAM) were used. Controls inoculated with a single bacterial culture in the corresponding medium were also analyzed. 

### 2.11. Statistical Analysis

All experiments were performed in duplicate or triplicate and carried out in a completely randomized design. Statistical analysis was performed using GraphPad Prism software 6.0. A one-way ANOVA was used for comparing data with one factor between groups. A two-way ANOVA was used for comparing more than one factor between groups, and differences were adjudicated using the post hoc analysis recommended by the software GraphPad Prism 6.0 (GraphPad Software Inc., San Diego, CA, USA).

## 3. Results

### 3.1. Isolation and Characterization of LAB

After the enrichment of microbes from diverse matrices, a total of 40 isolates grown on MRS agar were randomly selected and screened for antibacterial activity against *E. coli* O157:H7 CECT 4267 and *L. monocytogenes* CECT 4031. All isolates showed typical morphological characteristics, being Gram-positive bacilli and cocci; all the isolates were catalase-negative and non-motile. All of them showed a capacity to decrease the pH of the medium to 3.0, with tolerance to acidic and basic conditions (2.5 to 9). Out of the isolated microorganisms, only 24 isolates showed various levels of inhibition, from modest (ranging from 7 mm to 10 mm) to good (more than 10 mm) anti-*Listeria* and anti-*E. coli* activity (Table 1). Of these, only six strains with the best antibacterial activity were selected for further studies. The molecular analysis revealed that all the selected strains belonged to *Lactiplantibacillus plantarum*, with a percentage of homology higher than 98% (Table 2).

**Table 1 microorganisms-11-02679-t001:** Antibacterial activity of LAB isolated from various matrices against *Escherichia coli* O157:H7 CECT 4267 and *Listeria monocytogenes* CECT 4031, as determined using the agar diffusion method. Diameter halos inhibition expressed in mm. Mean values and standard deviations of three replicates are indicated. The six strains selected for further studies are underlined.

Isolated LAB Strain	Source/Matrix	*E. coli* (mm)	*L. monocytogenes* (mm)
** 3 nm **	Intestines of dead locusts	14.17 ± 0.29	13.17 ± 0.57
** 1 nm **	Intestines of dead locusts	15.00 ± 0.00	14.57 ± 2.32
** S4 **	Fermented green olive brine	15.00 ± 1.00	11.23 ± 0.68
** RSOLV **	Fermented olives	14.17 ± 0.76	13.43 ± 0.93
** pepp1 **	Fermented pepper brine	15.00 ± 0.00	15.10 ± 0.26
** pepp2 **	Fermented green pepper brine	15.00 ± 0.00	14.50 ± 0.50
**S5**	Horse sausage	10.00 ± 1.00	12.07 ± 0.40
**S6**	Horse sausage	0.00 ± 0.00	0.00 ± 0.00
**N8**	Dried anchovy	11.00 ± 1.00	0.00 ± 0.00
**N4c**	Dried fermented anchovy	12.00 ± 1.00	8.03 ± 0.25
**F1c**	Infant feces	9.13 ± 0.71	0.00 ± 0.00
**F5a**	Infant feces	12.90 ± 0.17	9.97 ± 0.15
**LM**	Breast milk	9.83 ± 0.29	9.92 ± 0.14
**F1**	Infant feces	10.17 ± 0.29	11.00 ± 0.00
**Rg4a**	Artisanal Tunisian ricotta cheese	10.17 ± 0.29	11.17 ± 1.26
**RL4**	Tunisian fermented milk Leben	8.83 ± 0.29	12.93 ± 0.12
**K10**	Tunisian artisanal Gueddid	0.00 ± 0.00	10.60 ± 0.36
**LC4**	Goat milk	0.00 ± 0.00	10.50 ± 0.50
**S1**	Fermented olive brine	10.80 ± 0.36	10.47 ± 0.50
**AIB**	Rabbit intestines	0.00 ± 0.00	11.00 ± 0.00
**S5**	Horse sausage	11.00 ± 1.00	11.00 ± 0.87
**N8**	Dried anchovy	9.90 ± 0.10	9.00 ± 0.00
**K10**	Tunisian Gueddid	0.00 ± 0.00	0.00 ± 0.00
**O3**	Fermented olive brine	0.00 ± 0.00	0.00 ± 0.00

The growth of the six selected strains was also evaluated in different conditions. The tolerance to acidic (pH = 4.2) and alkali (pH = 9.2) pH was evaluated at different concentrations of salts NaCl (2%, 4%, and 8%, *w*/*v*). The ability to grow in MRS medium with different types and different proportions of sugar was also evaluated, e.g., glucose, fructose, and sucrose in 2% and 4%, respectively. All the strains were viable under all the conditions tested and could survive at different pH values, as well as tolerate up to 8% NaCl (Table 3).

### 3.2. Antimicrobial Activity and Some Insights into the Antibacterial Agents

All the selected strains showed antimicrobial activity against the bacterial indicators, with various inhibition zones ranging between 11 mm and 15 mm (Table 1). Four of them, namely, 1 nm, S4, pepp1, and pepp2, demonstrated a higher antimicrobial activity against *E. coli*. In order to determine the nature of the antibacterial substance, the CFS of each LAB strain was collected after 24 h, and the antibacterial activity of the crude CFS was compared with that of the corresponding crude cells, evidencing that the inhibitory activity depended on secreted substances (Appendix A). For the six strains, neutralizing the CFS with NaOH abolished the antimicrobial activity with little or no impact on the growth of indicator bacteria, thus suggesting that the inhibitory effects were due to organic acids (Table 4 and Table 5). Moreover, heat treatment of the CFS did not suppress their antimicrobial effect, indicating the presence of thermostable antimicrobial substances. Crude cells and heat-treated cells did not exhibit any antimicrobial effect, confirming the external secretion of the antibacterial agent and not attachment to the cell membrane. 

Quantification of the organic acid contents in the CFS from the selected *Lpb. plantarum* strains are reported in Table 6. All the strains were able to produce various organic acids, and lactic acid was the most abundant, with a concentration above 13 g/L. All the strains, except RSOLV and pepp2, were able to produce ascorbic acid. Only the strain 3 nm was able to produce fumaric acid with a quantity of 15.47 ± 6.71 mg/L. These findings confirm the heterofermentative aspect of *Lpb. plantarum* strains and suggest that the combination of organic acids could be the reason for the observed antimicrobial activity.

Antifungal activities of the six LAB strains against *P. expansum* CECT 2278, *A. niger* CECT 2805, *F. culmorum* CECT 2148, *S. cerevisiae*, and *B. cinerea* CECT 20973 were assessed, and all the strains were active against the four indicator fungal species. All the strains were able to inhibit *P. expansum* and *F. culmorum*, with inhibition halos ranging between 4 mm and 8 mm (Table 7). Only two strains (RSOLV and pepp2) showed the best antagonistic effect toward *B. cinerea*. Only one strain, namely, RSOLV, was weakly active against *A. niger* compared with the antifungal activity of the others (Appendix A). None of the LAB strains tested in this study showed inhibitory activity against *S. cerevisiae*. 

### 3.3. Auto-Aggregation and Co-Aggregation Assays

Auto-aggregation and co-aggregation abilities are related to the adherence capability to intestinal epithelial cells. Moreover, it is an important phenotypic trait for screening potential probiotic strains. The obtained results showed that all the selected LAB strains exhibited varied auto-aggregation indices, which were time-dependent (Table 8). Indeed, the auto-aggregation of the LAB strains ranged from 28.87% ± 3.68% to 16.70% ± 1.12% and from 70.30% ± 4.73% to 34.67% ± 0.99% after 4 h and 24 h, respectively. Regarding the auto-aggregation abilities after 4 h, the highest auto-aggregation capacities were observed for the *Lpb. plantarum* strain isolated from fermented olives (RSOLV: 28.87% ± 3.68%), and the lowest auto-aggregation index was obtained for the *Lpb. plantarum* strain isolated from locusts (1 nm) with a value of 16.70% ± 1.12%. After 24 h, the highest values of auto-aggregation capability were obtained from the fermented olive strain S4 as well as the fermented pepper strain pepp2 with a value of 70.30% ± 4.73% and 69.82% ± 2.20%, respectively. It is noteworthy that *Lpb. plantarum* pepp2 showed the highest Δ auto-aggregation index with a value of 53.12 after 20 h of incubation. 

Co-aggregation ability with *Listeria* was shown by all LAB, which increased over the tested times (4 h and 24 h) at pH = 7.0. The co-aggregation capacity ranged from 7.94% ± 0.94% to 4.99% ± 0.55% and from 8.76% ± 1.16% to 4.61% ± 0.79% after 4 h and 24 h, respectively (Table 9). Regarding the co-aggregation abilities after 24 h, the highest co-aggregation capacities were observed for *Lpb. plantarum* RSOLV (8.76% ± 1.16%) and S4 (7.37% ± 0.46%); both of these strains were isolated from fermented olives. 

The co-aggregation capacity to *E. coli* ranged from 28.30% ± 10.7% to 15.05% ± 0.89% and from 54.88% ± 6.98% to 36.96% ± 5.72% after 4 h and 24 h, respectively, as shown in Table 8. Regarding the co-aggregation abilities after 24 h, the highest co-aggregation capacities were observed for the *Lpb. plantarum* 1 nm strain (54.88% ± 6.98%), followed by *Lpb. plantarum* strains isolated from fermented vegetables RSOLV (51.15% ± 4.69%). These results showed that the co-aggregation ability of *Lpb. plantarum* strains were strain-specific, with the highest co-aggregation capacity toward *E. coli* in comparison with *L. monocytogenes*.

### 3.4. Safety Features: Antibiotic Resistance and Hemolytic Activity

Antibiotic (ATB) susceptibility is one of the required properties by which specific strains can be considered a potential probiotic bacterium. Indeed, probiotics should not harbor acquired antibiotic resistances. The six selected *Lbp. plantarum* strains were phenotypically analyzed toward various types of ATBs, as reported below (Table 10). Most of the strains were resistant toward kanamycin, streptomycin, and tetracyclin while being sensitive and intermediate to ampicillin and gentamycin. All strains were considered non-hemolytic (γ-hemolysis). 

### 3.5. In Vitro Oro-Gastrointestinal Transit Tolerance Assay

The selected *Lpb. plantarum* strains were investigated for their capability to tolerate OGI stress in vitro (Figure 3). To mimic the OGI transit condition, bacterial cultures were exposed sequentially to lysozyme, pepsin, and acidic pH and then bile salts and pancreatic enzymes. As shown in Figure 4, all the strains were able to survive until the end of the experiment and with different levels of viability. There was no significant reduction in the survival rate in the first two steps (i.e., saliva stress and gastric stress at pH 3) compared to the control; however, a significant decrease in viability was observed for all the strains after incubating at pH 2 in the presence of pepsin. In fact, the viability index decreased four times in comparison with the first two steps. All the strains showed a similar tolerance level of the positive control *L. reuteri* DSM 17938 with a value of 3.80% ± 0.75%. Nevertheless, the survival rate decreased slightly under simulated intestinal conditions (bile salts and pancreatic enzymes). The highest cell viability was detected for *Lpb. plantarum* isolated from fermented olive S4 (2.33 ± 0.4), followed by locust strains 1 nm (1.90 ± 0.5). 

### 3.6. Biofilm Formation

Biofilm formation abilities on the abiotic surface were investigated on 96-well polystyrene microliter plates after 24 h, 48 h, and 72 h using two staining methods: resazurin, which allows for evaluating the viability of adherent bacterial cells, and crystal violet, which stains both viable and not viable adherent cells (Figure 4).

Regarding resazurin detection (Figure 4A), the six *Lbp. plantarum* strains were found to be weak to moderate biofilm producers. In addition, their absorbance was similar to the positive control *E. faecalis* V583. Based on the absorbance, after 24 h *Lpb. plantarum* S4, isolated from fermented olives, showed the highest rate of adherent viable cells (0.22 ± 0.02). After 72 h, the highest rate of viable cells was recorded for *Lpb. plantarum* isolated from locusts, namely, 1 nm, with a value of 0.29 ± 0.00. No significant difference was observed between 24 h and 48 h of incubation for all the strains.

The biofilm formation was also investigated using a crystal violet assay (Figure 4B). Similar to what was found with resazurin, all tested strains were able to form biofilm. Based on absorbance values, *Lpb. plantarum* S4, isolated from fermented olives, was found to be the best biofilm producer candidate after 24 h. A significant difference was confirmed (*p* < 0.001) between 24 h and 72 h of incubation for all the strains. On the other hand, pepp2 showed a good ability to form biofilm on abiotic surfaces after 48 h at 37 °C. The highest rate of biofilm production after 72 h was recorded for the same strain, *Lpb. plantarum* S4, with a value 1.55 ± 0.01.

### 3.7. Cytotoxic Effects

The cytotoxicity of CFS from the analyzed *Lbp. plantarum* strain was evaluated on the cancer cell LS line upon incubation (24 h and 72 h) at different concentrations (Figure 5); the bacterial medium (MRS) was used as a solvent (negative) control and did not show any cytotoxic effect.

All the strains showed an inhibitory effect on the LS cell line after 24 h, with IC_50_ ranging between 7.48 mg/mL and 12.06 mg/mL (Table 11). The results showed that the best candidate was 1 nm isolated from locust intestines with IC_50_ = 7.48 mg/mL. Regarding the anti-proliferative activity of CFSs on LS cells, the data showed that toxicity depends on the incubation duration. Furthermore, after 72 h, IC_50_ ranged from 2.18 mg/mL to 5.06 mg/mL, corresponding, respectively, to pepp2 and 3 nm strains. At a concentration of 8 mg/mL of CFS and after 72 h of incubation, the viability of LS cells decreased with values ranging between 33.64% and 18.68%, corresponding, respectively, to 1 nm CFS and pepp1 CFS. After 24 h and 72 h, starting from the concentration of 1 mg/mL of CFS, all the strains exhibited a significant inhibition (*p* < 0.0001) of LS proliferation, and at a concentration of 10 mg/mL, the percentage of surviving cells was reduced by 50%. The strains 1 nm and S4, isolated from locust intestines and fermented olives, respectively, showed a higher significant toxicity effect compared with the other strains in all tested concentrations.

### 3.8. Adhesion to Human Enterocyte-Like Cells

In this study, the adhesive capacities of the selected *Lpb. plantarum* strains were assessed and compared to that of commercial *Lpb. plantarum* 299 V, a probiotic model. The obtained data confirm the adhesive potential of all the strains, which were higher or equal to the adhesion value of *Lpb. plantarum* 299 V (2.5 ± 0.3%), ranging between 3.4% ± 2.5% and 6.2% ± 2.5% (Figure 6). The strains *Lpb. plantarum* 3 nm (6.2 ± 2.4, *p* < 0.05) and S4 (5.5 ± 1.4%) exhibited the highest adhesion percentage with a value of more than 5%, although the difference between S4 and *Lpb. plantarum* 299 V was not significant. No significant difference was found between the tested *Lpb. plantarum*.

### 3.9. Application in Food Model

#### 3.9.1. Adhesion on the Tomato Surface

The adhesion of the investigated *Lpb. plantarum* strains to the tomato surface was performed using grape tomatoes (*Solanum lycopersicum L*.) purchased from a local market. The survival and adhesion rates were measured after 2, 3, and 5 days of incubation at ambient temperature, respectively (Figure 7). All the strains could survive and attach to the tomato surface after 5 days of incubation at ambient temperature with different levels. The strain *Lpb. plantarum* 1 nm showed the best performance of adhesion to tomatoes after 5 days, contrary to strain RSOLV, which showed a decreased level of attachment during the incubation period. After 3 days of incubation, all the strains showed no significant difference (*p* < 0.05) with a range of attachment ranging between 4.02 and 5.96 (LogCFU/g tomato). Comparing the capacity to adhere to tomatoes as a function of incubation time, the strain *Lpb. plantarum* pepp2 showed a lower rise in capacity with delta time Δt = 0.728 between the third and fifth days of incubation.

#### 3.9.2. Biocontrol/Pathogen Antagonism on Tomatoes

In order to elucidate the interaction between LAB and pathogens on a model food matrix, each of the investigated *Lpb. plantarum* strains were mixed with a bacterial pathogen and then inoculated on the tomatoes (Table 12). The tomatoes were kept for 5 days at room temperature to mimic ordinary conditions; later, the attached and alive microorganisms were counted on the selective medium MRS agar, SMAC agar, and PALCAM agar. Bacteria (LAB, *E. coli*, and *L. monocytogenes*) inoculated in pure cultures were used as a control. It seemed that the ability to adhere to the tomatoes was affected by time (Table 11). Regarding the co-cultures, all LAB strains showed a significant decrease in attachment (*p* < 0.0001) after 3 days of incubation, ranging from 1.78 ± 1.53 Log CFU/g tomato to 1.42 ± 1.26 Log CFU/g tomato followed by an increase on the fifth day, and only 1 nm remained stable compared with the other strains. In fact, all *Lpb. plantarum* strains co-cultured with *E. coli* or *L. monocytogenes* were able to attach and survive on the tomatoes after 5 days and at a similar rate to the control (Appendix A). The results indicated that the *Lpb. plantarum* strains were able to significantly decrease/inhibit the viability of *E. coli* on the fifth day with a range of 1.38 ± 0.31–1.13 ± 0.45 Log CFU/g tomato after the first day of incubation. The inhibitory effect on *E. coli* was more pronounced in the presence of the strain *Lpb. plantarum* S4, while the strain 3 nm showed the best survival rate in co-culture. In the co-cultures (*Lpb. plantarum*-*Listeria*), an inhibitory effect on the growth of the pathogen was observed after 3 days in the presence of each of the six *Lpb. plantarum* strains, with the *L. monocytogenes* count ranging between 1.00 ± 0.67 and 1.54 ± 1.41 Log CFU/g tomato. On the fifth day, the CFU number of *Listeria* in co-culture with *Lpb. plantarum* was significantly reduced compared with the control. Interestingly, strains 1 nm and S4 inhibited completely (*p* < 0.05) the growth of *L. monocytogenes* on tomatoes after 5 days.

## 4. Discussion

Probiotics have been traditionally isolated from dairy products, though, in recent years, there has been an increasing trend in exploring novel and alternative sources [61]. Unconventional niches that are being investigated to discover new probiotics include traditional fermented foods and beverages, vegetables, fruit juices [62,63], and the intestine of insects [64,65]. According to scientific research, fermented vegetables are interesting sources of potentially probiotic LAB strains [7,66,67]. Many Tunisian publications reported that fermented vegetables host LAB strains with an antimicrobial and bio-protective effect against germs and moulds [68,69,70], but only a few articles mentioned the possible probiotic properties of strains from Tunisian fermented vegetables. 

The present study aimed to isolate LAB strains from unconventional sources and to characterize them for further usage, such as probiotic cultures and/or for food biocontrol. After screening presumptive LAB from very diverse sources, including infant feces, breast milk, rabbit intestine, fermented olive, etc., we focused on six strains exhibiting the highest antagonistic action toward foodborne pathogens. Such selected strains, derived from fermented vegetables (fermented olives and peppers) and locust intestines, were identified as *Lactiplantibacillus plantarum* and were further investigated for their antimicrobial, probiotic, and biocontrolling properties. Our findings are supported by similar works that reported *Lpb. plantarum* as the dominant group in fermented vegetables, e.g., tomatoes and table olives [71,72]. According to recent studies, *Clostridium* (*Firmicutes*) is one of the most abundant genera found in insects [65]. Likewise, Garofalo et al. [73] confirmed the dominance of *Clostridium* in locusts, mainly represented by *Enterobacteriaceae* and *Weissella* spp. Another study reported the presence of *Lactobacillus metriopterae* sp. nov. in locust guts [74]. Unexpectedly, in the current study, LAB isolated from locusts were identified as *Lpb. plantarum* with similarity over 98%, and we can mention that these are the first *Lpb. plantarum* found and characterized in locusts. LAB isolated from insects are considered promising probiotics for the benefit of human and animal health due to the survival/persistence of their host in hard environments [65]. Indeed, some very recent research has looked into the probiotic potential of LAB obtained from insect guts [64,75,76]. A complete genome sequence of *Weissella confusa* LM1, found in the gut of the migratory locust, indicated the ability to adapt to different ecological niches [77]. While there are studies on microbiological communities (including LAB) in grasshoppers (*Locusta migratoria migratorioides*) sold for human consumption [78], to our knowledge, this is the first report studying the probiotic potentialities of locust-derived LAB.

Nowadays, probiotics with antimicrobial activity are becoming an alternative to traditional drugs due to antibiotic resistance diffusion [79]. Interestingly, it was found that all the strains were able to inhibit the indicator’s growth; a similar study [80] showed that *Lpb. plantarum* exhibits an inhibitory effect against *L. monocytogenes* and *E. coli* similar to our tested LAB. Moreover, a recent study confirmed the antibacterial potential of *Lpb. plantarum* insect strains with a similar range against *E. coli* [64]. 

The CFSs from the investigated *Lpb. plantarum* strains had pH values in the range of 3.5, as reported by previous publications [81]. After pH neutralization to 6.0, all CFSs showed minimal to no activity against all the pathogens tested, proving the role of organic acids in antimicrobial effects. It was reported that the increased production of organic acid through carbohydrate fermentation decreases the pH of the medium, which is the major factor suppressing pathogen growth [82]. In a similar study [83], CFSs from LAB, including *Lpb. plantarum*, showed anti-*E. coli* activity with inhibition zones ranging between 12.89 ± 0.21 and 15.32 ± 0.28 mm. The authors confirmed that the antimicrobial activity was due to the combination of various metabolites, including organic acids. In fact, CFSs of LAB are a complex mixture of metabolic enzymes, secreted proteins, short-chain fatty acids, vitamins, amino acids, peptides, organic acids, and cell components [84]. It was recently reported that *Lpb. plantarum* is able to synthesize various beneficial extracellular metabolites, known as postbiotics [20], i.e., bioactive soluble compounds or peptides that are produced during LAB growth [85] and confer health benefits such as infection prevention [86] and antitumor and immunomodulatory effects [87]. Similarly, previous works have demonstrated the importance of organic acids as bio-preserving agents [88,89]. In addition, Mirzaei et al. [90] reported that the antimicrobial activity of LAB strains disappeared when their CFSs were adjusted to pH 6.5 and treated with catalase. Nevertheless, the bacteriocin production by LAB is highly affected by several factors, including temperature, pH, and incubation time. It was also reported that the optimum secretion of bacteriocin is when the pH ranges between 5.0 and 6.0 [91]. In the present study, all the tested strains showed antibacterial activity against the target pathogens, most probably due to the organic acids secreted in their CFSs.

The antifungal potential of LAB strains was also estimated, and our results showed that LAB inhibit the growth of *P. expansum*, *A. niger*, *F. culmorum*, and *B. cinerea*. These findings are similar to the work carried out by other investigators [92,93]. A recent Turkish publication [94] showed that *Lpb. plantarum* was active against *P. expansum* and *A. niger*. Indeed, several studies reported that LAB isolated from vegetables and plants possess a better antifungal activity [95,96], while LAB derived from dairy products exhibit antibacterial activity against foodborne pathogens through bacteriocin production capabilities [97,98]. 

Thereafter, the probiotic traits were investigated. Starting with OGI transit tolerance, all the tested *Lpb. plantarum* strains were incubated in successive solutions to mimic the human OGI transit, and their viability was evaluated. In fact, to be applied in the food industry, LAB should possess good resistance to acidic environments, especially in the preparation of high-acid foods [99]. All selected *Lpb. plantarum* strains were able to survive the simulated OGI transit with a value ranging between 2.33 ± 0.4 and 1.41 ± 0.44 Log CFU/mL, with the LAB strain of fermented olive S4 exhibiting the highest resistance. Moreover, all the strains showed good resistance to acidic pH through OGI transit, which is congruent with several findings in the literature. Indeed, *Lpb. plantarum* was proven to be able to survive in pH varying between 2.5 and 4 [100,101]. Furthermore, in agreement with our findings, several authors demonstrated the ability of *Lpb. plantarum* to survive to OGI transit with a rate of 3 Log CFU/mL [43,102].

We further examined the potential probiotics for cell-binding properties. In the current study, all tested LAB exhibited time-dependent auto-aggregation ability; particularly, the fermented olive-derived strains (RSOLV and S4) showed the best auto-aggregation performance. In fact, the auto-aggregation of bacteria has been associated with adherence ability to intestinal cells, a prerequisite for the colonization of the gastrointestinal tract [103]. In a similar earlier study [104], *Lpb. plantarum* isolated from fermented vegetables showed auto-aggregation rates very close to our results.

In the co-aggregation assays, using the same foodborne pathogenic bacteria as for the antibacterial assay, the co-aggregation ability of all tested LAB increased over time, and it was high for *Lbp. plantarum* strains from fermented green pepper, indicating a potential to prevent and/or exclude colonization of pathogens in the gastrointestinal tract. The adhesion ability to cells was strain-specific, as it varied even within the same species. Our findings are supported by recent work on LAB strains that obtained similar results [46,60] and are in agreement with the study by Ben Taheur et al. [68], which reported a lower rate of co-aggregation of probiotic LAB with *E. coli*. 

Regarding adhesion capability, the *Lpb. plantarum* strains were first evaluated for biofilm production ability on polystyrenes surface, then for adhesion to human enterocyte-like cells, and, finally, to tomatoes as a food model. In fact, bacterial adhesion to epithelial cells is considered one of the most accurate features for selection criteria for probiotic strains [105]. The results showed that all selected strains were able to form biofilm on abiotic surfaces, especially S4 and 1 nm, for a longer time, with a value similar to previous studies [106], demonstrating that *Lpb. plantarum* strains provide various levels of biofilm production capability from “no biofilm” to “good biofilm producer”, possibly depending on the amount of exopolysaccharides produced [107]. Furthermore, knowledge about the surface conditions and the bacterial properties influencing adhesion is still insufficient [108]. Concerning the adhesion to human cells, our findings demonstrated that all the tested *Lpb. plantarum* strains survived and attached to Caco-2 cells better than the probiotic model strain used as control. Our results are comparable to values previously obtained for *Lpb. plantarum* [43]. We also found that the CFSs of the tested *Lpb. plantarum* strains were able to decrease colorectal adenocarcinoma cell viability, proving their anti-cancer and anti-proliferative potential. These findings are in line with current studies reporting the cytotoxicity of *Lpb. plantarum* metabolites on cancer cells [20,55]. 

In the present study, we suggest the use of *Lpb. plantarum* strains as probiotics and bio-controlling agents on tomatoes. Non-dairy, plant-based food matrices such as fruits, vegetables, and legumes have been used successfully to produce probiotic products [109]. The investigated *Lpb. plantarum* strains showed strong attachment in vivo to tomatoes. Arellano-Ayala et al. [60] reported similar adhesion percentages of LAB to tomato fruits. The selection of tomatoes as a model to elucidate the adhesion ability of the probiotic strains reflects the importance of this vegetable in the human diet. The latest research investigated LAB’s efficiency as a protective culture tool to control *L. monocytogenes* in ready-to-eat and dairy-ripened products [110]. Moreover, Yin et al. confirmed the possible use of LAB as biocontrol agents for limiting/inhibiting pathogens contamination on leafy greens [32]. Here, we investigated the ability of *Lpb. plantarum* strains to attach to tomatoes in co-culture with pathogens. The enumeration of microorganisms was performed in different selective mediums; the biocontrol assay on tomatoes confirmed the ability of all *Lpb. plantarum* strains, principally 1 nm and S4, to antagonize the tested pathogens, with a maintained count of CFU after 5 days and with a significant difference compared with the control. 

In conclusion, this is one of the few studies that have investigated the probiotic potential of Tunisian vegetables- and locust intestine-derived LAB and their biocontrol capacities on a food model matrix. Six selected *Lpb. plantarum* strains were analyzed for their antagonism potential, demonstrating that they could be good candidates as food-protective cultures and with an interesting probiotic profile. Nevertheless, further studies are needed to deepen their characterization and to guarantee their use as probiotic strains in biocontrol.

## Figures and Tables

**Figure 1 microorganisms-11-02679-f001:**
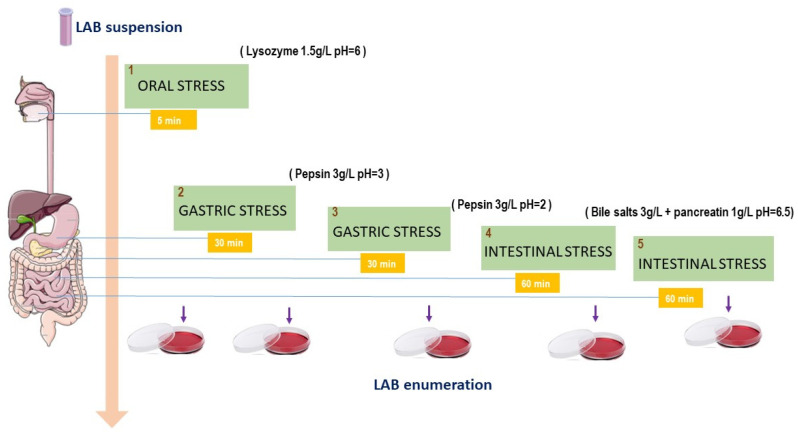
Schematic diagram of the oro-gastrointestinal transit (OGI) assay used to test the tolerance of *Lpb. plantarum* strains in vitro. Bacterial cultures were first washed twice with sterile buffer and then incubated in lysozyme solution to mimic the saliva stress for 5 min. Later, the pellet was incubated in a solution containing pepsin with different pH values, arriving at intestinal stress, in which cells were inoculated in bile salts and pancreatin for 60 min and then incubated for 1 h at 37 °C in a sterile electrolyte solution.

**Figure 2 microorganisms-11-02679-f002:**
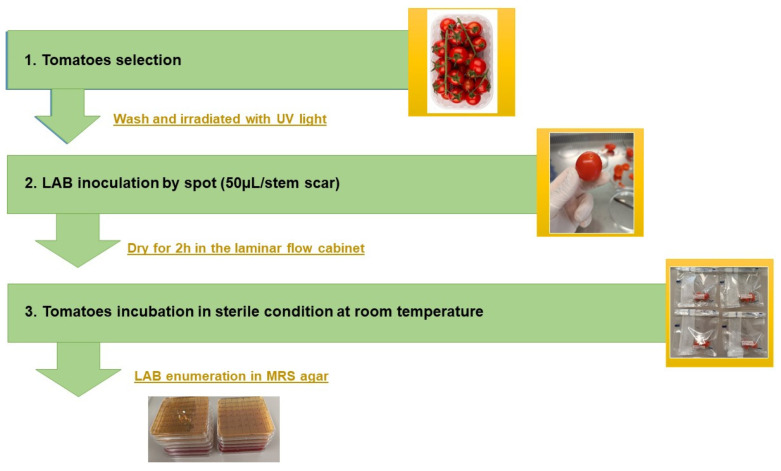
Schematic representation of adhesion in the food model assay. First, tomatoes were washed with sterile distilled water and decontaminated with UV light for 15 min; then, a scar was created using a sterile syringe and *Lpb. plantarum* cultures were inoculated in spots. Inoculated tomatoes were left for 2 h to dry under a laminar flow hood and incubated at room temperature. Sterile PBS buffer pH = 7.0 was used as a control.

**Figure 3 microorganisms-11-02679-f003:**
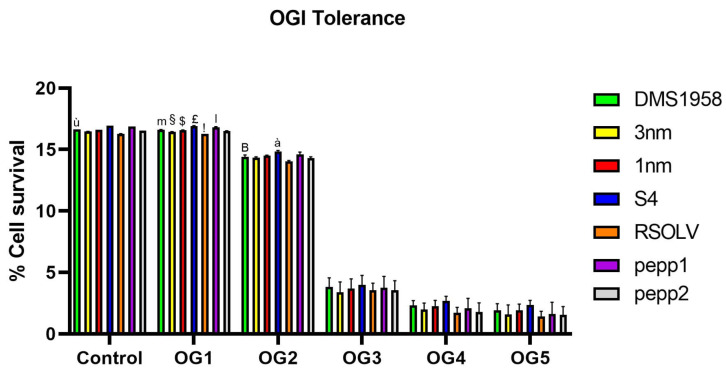
Oro-gastrointestinal tolerance in vitro shown by *Lpb. plantarum* strains. Strains were incubated in MRS (control) and then exposed successively to lysozyme (OG1), pepsin pH 3 (OG2), pepsin pH 2 (OG3), and bile salts and pancreatic enzymes (OG4). Finally, bacterial cultures were incubated in an intestinal electrolyte solution pH 6.5 (OG5). The assay was performed in triplicate. ANOVA was used to identify significant differences at each time point (control, OG1, OG2, OG3, and OG4) of the OGI transit between strains. ^ù^ *p* < 0.0001 vs. all strains; ^m^ *p* < 0.05 vs. OG1 of strains 3 nm, S4, RSOLV, pepp1, and pepp2; ^§^ *p* < 0.05 vs. OG1 of strains 1 nm, S4, RSOLV, and pepp1; ^$^ *p* < 0.0001 vs. OG1 of strains S4, RSOLV, and pepp1; ^£^ *p* < 0.0001 vs. OG1 of strains RSOLV and pepp2; ^!^ *p* < 0.0001 vs. OG1 of strains pepp1 and pepp2; ^l^ *p* < 0.0001 vs. OG1 pepp2; ^B^ *p* < 0.05 vs. OG2 of strains S4 and RSOLV; ^à^ *p* < 0.05 vs. OG2 of strains 3 nm, 1 nm, RSOLV, and pepp2.

**Figure 4 microorganisms-11-02679-f004:**
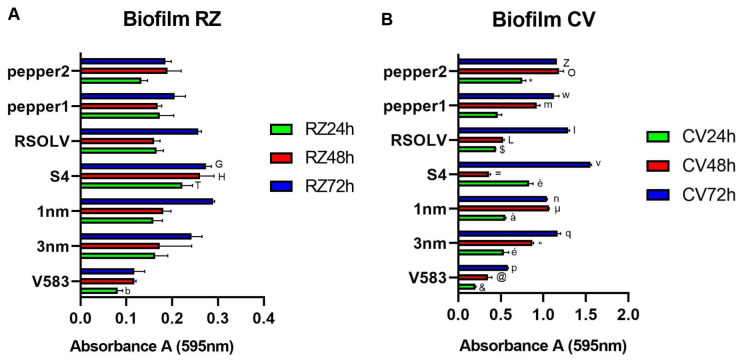
Biofilm formation (expressed as absorbance at 595 nm) of the *Lpb. plantarum* strains after 24 h and 72 h of incubation at 37 °C using resazurin (RZ) (**A**) or crystal violet (CV) (**B**). Error bars indicate the standard deviation of triplicate experiments. ANOVA was used between strains and to control at each time (24 h, 48 h, and 72 h).^b^ *p* < 0.05 vs. RZ 24 h of all strains except pepp2; ^T^ *p* < 0.05 vs. RZ 24 of strains 3 nm, 1 nm, and pepp2; ^H^ *p* < 0.05 vs. RZ 24 h of strains V583, RSOLV, and pepp1; ^G^ *p* < 0.05 vs. RZ 72 h of strains V583, RSOLV, and pepp1; ^&^ *p*  <  0.0001 vs. CV 24 h of all strains; ^é^ *p* < 0.0001 vs. CV 24 h of strains S4 and pepp2; ^à^ *p* < 0.05 vs. CV 24 h of strains S4, RSOLV, and pepp2; ^è^ *p* < 0.0001 vs. CV 24 h of strains RSOLV and pepp1; ^$^ *p* < 0.001 vs. CV 24 h of strain pepp2; * *p* < 0.0001 vs. CV 24 h of strain pepp1; @ *p* < 0.0001 vs. CV 48 h of all strains except S4; “ *p* < 0.0001 vs. CV 48 h of all strains except pepp1; ^µ^ *p* < 0.0001 vs. CV 48 h of all strains; ^=^ *p* < 0.0001 vs. CV 48 h of all strains; ^L^ *p* < 0.0001 vs. CV 48 h of all strains; ^m^ *p* < 0.0001 vs. CV 48 h of all strains; ^O^ *p* < 0.0001 vs. CV 48 h of all strains; ^p^ *p* < 0.0001 vs. CV 72 h of all strains; ^q^ *p* < 0.0001 vs. CV 72 h of all strains except pepp1 and pepp2; ^n^ *p* < 0.05 vs. CV 72 h of all strains; ^v^ *p* < 0.0001 vs. CV 72 h of all strains; ^I^ *p* < 0.0001 vs. CV 72 h of all strains; ^w^ *p* < 0.0001 vs. CV 72 h of strains 1 nm, S4, and RSOLV; ^Z^ *p* < 0.05 vs. CV 72 h of strains 1 nm, S4, and RSOLV.

**Figure 5 microorganisms-11-02679-f005:**
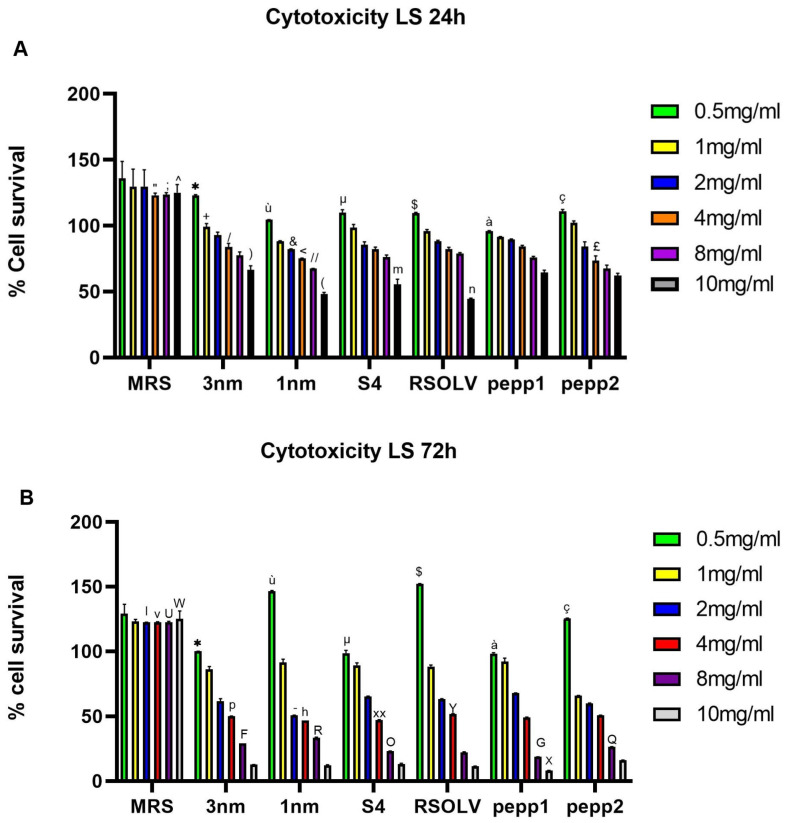
The effect of different concentrations of CFS of *Lpb. plantarum* strains on the proliferation of LS cells after 24 h (**A**) and 72 h (**B**) incubation at 37 °C. Error bars indicate the standard deviation of triplicate experiments. A one-way ANOVA test was used at each concentration to compare strains among concentrations and to the control (MRS). * *p* < 0.05 vs. all 3 nm and all MRS doses; ^ù^ *p* < 0.05 vs. all 1 nm and all MRS doses; ^µ^ *p* < 0.05 vs. all S4 and all doses; ^$^ *p* < 0.05 vs. all RSOLV and all MRS doses; ^à^ *p* < 0.005 vs. all pepp1 and all MRS doses; ^ç^ *p* < 0.05 vs. all pepp2 and all MRS doses. ^+^ *p* < 0.0001 vs. 1 mg/mL MRS; ^&^ *p* < 0.0001 vs. 2 mg/mL MRS; ^“^ *p* < 0.0001 vs. 4 mg/mL of all strains; ^£^ *p* < 0.05 vs. 4 mg/mL of the strains 3 nm, S4, RSOLV, and pepp1; ^/^ *p* < 0.05 vs. 4 mg/mL of the strain 1 nm; ^<^ *p* < 0.05 vs. 4 mg/mL of strains S4, RSOLV, and pepp1; ^;^ *p* < 0.0001 vs. 8 mg/mL of all strains; ^//^ *p* < 0.05 vs. 8 mg/mL of strains 3 nm, S4, RSOLV, and pepp1; ^ *p* < 0.0001 vs. 10 mg/mL of all strains); *p* < 0.05 vs. 10 mg/mL of the strains 1 nm, S4, and RSOLV; (*p* < 0.05 vs. 10 mg/mL of the strains pepp1 and pepp2; ^m^ *p* < 0.05 vs. 10 mg/mL of the strains RSOLV and pepp1; ^n^ *p* < 0.05 vs. 10 mg/mL of the strains pepp1 and pepp2; ^l^ *p* < 0.05 vs. 2 mg/mL of all strains; ^-^ *p* < 0.05 vs. 2 mg/mL of the strains S4 and pepp1; ^v^ *p* < 0.05 vs. 4 mg/mL of all strains; ^p^ *p* < 0.05 vs. 4 mg/mL of the strains 1 nm, S4, RSOLV; ^h^ *p* < 0.05 vs. 4 mg/mL of the strains RSOLV, pepp1 and pepp2; ^XX^ *p* < 0.05 vs. 4 mg/mL of all strains; ^Y^ *p* < 0.05 vs. 4 mg/mL of the strain pepp1; ^U^ *p* < 0.001 vs. 8 mg/mL of all strains; ^F^ *p* < 0.05 vs. 8 mg/mL of all strains; ^R^ *p* < 0.05 vs. 8 mg/mL of all strains; ^O^ *p* < 0.05 vs. 4 mg/mL of the strains 3 nm, 1 nm, pepp1, and pepp2; ^G^ *p* < 0.05 vs. 8 mg/mL of all strains; ^Q^ *p* < 0.05 vs. 8 mg/mL of all strains; ^W^ *p* < 0.001 vs. 10 mg/mL of all strains; ^X^ *p* < 0.05 vs. 10 mg/mL of pepp2.

**Figure 6 microorganisms-11-02679-f006:**
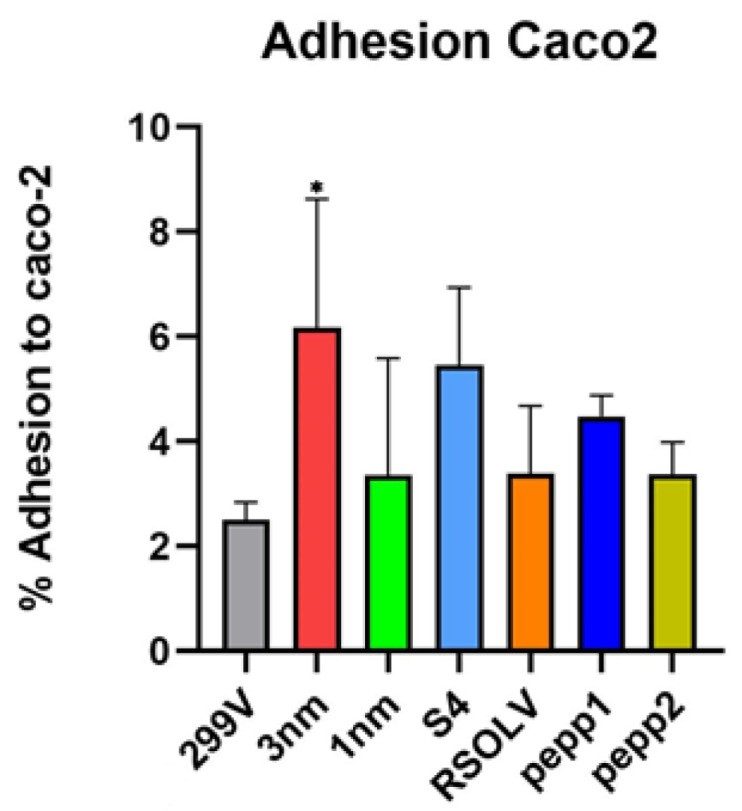
Adhesion percentage of *Lpb. plantarum* strains to Caco-2 monolayers after 1 h of co-incubation at 37 °C. *Lpb. plantarum* V299 was used as control. Error bars indicate the standard deviation of triplicate experiments. *: *p* < 0.05 vs. *Lpb. plantarum* 299 V.

**Figure 7 microorganisms-11-02679-f007:**
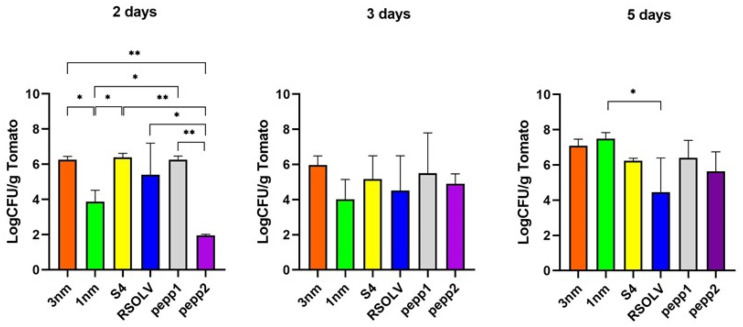
Adhesion percentage/capacity of *Lpb. plantarum* strains on tomatoes after 2, 3, and 5 days. Error bars indicate the standard deviation of triplicate experiments. A one-way ANOVA test was performed for strains at each incubation time (2 days, 3 days, and 5 days) with Tukey’s multiple comparisons test, * *p* < 0.05; ** *p* < 0.0001.

**Table 2 microorganisms-11-02679-t002:** The closest species/strain to the identified isolates according to 16S rRNA sequences.

Isolate	Source	Closest Species/Strain	Percentage of Identity	Accession Number *
**3 nm**	Dead locust (intestines)	*Lactiplantibacillus plantarum*	100%	OR431596
**1 nm**	Dead locust (intestines)	*Lactiplantibacillus plantarum*	99.86%	OR431597
**S4**	Fermented olive	*Lactiplantibacillus plantarum*	99.91%	OR431698
**RSOLV**	Fermented olive	*Lactiplantibacillus plantarum*	99.93%	OR431599
**pepp1**	Fermented pepper	*Lactiplantibacillus plantarum*	99.82%	OR431600
**pepp2**	Fermented pepper	*Lactiplantibacillus plantarum*	100%	OR431601

* Accession number of the sequences of the bacterial isolates in this study that were deposited in the NCBI database.

**Table 3 microorganisms-11-02679-t003:** Physiological and biochemical characteristics of the selected *Lpb. plantarum* strains after growth in different conditions: osmotic stress, acidic stress, and growth in different sources of carbon e.g., glucose (Glc), fructose (Fruc), and sucrose (Sucr).

Experimental Assay	3 nm	1 nm	S4	RSOLV	pepp1	Pepp2
Mobility	-	-	-	-	-	-
2% NaCl growth	+	+	+	+	+	+
4% NaCl growth	+	+	+	+	+	+
8% NaCl growth	+	+	+	+	+	+
MRS.Glc 2%	+	+	+	+	+	+
MRS.Glc 4%	+	+	+	+	+	+
MRS.Fruc 2%	+	+	+	+	+	+
MRS.Fruc 4%	+	+	+	+	+	+
MRS.Sucr 2%	+	+	+	+	+	+
MRS.Sucr 4%	+	+	+	+	+	+
MRS pH 3	+	+	+	+	+	+
MRS pH 4	+	+	+	+	+	+
MRS pH 5	+	+	+	+	+	+
MRS pH 9.2	+	+	+	+	+	+

**Table 4 microorganisms-11-02679-t004:** Antibacterial activity of whole cultures, CFS, and cells from the selected *Lpb. plantarum* strains. The CFS was used crudely or after neutralization with NaOH; *Lpb. plantarum* cells were tested before and after heat treatment. The indicator bacteria was *Escherichia coli* CECT 4267. No (-) or strong (++) inhibition showed a zone lower than 7 mm, ranging from 7 to 10 mm, or more than 10 mm, respectively. Assays were performed in duplicate.

	*E. coli*					
Strain	LAB Culture	Crude CFS	pH-Neutralized CFS	Temperature-Treated CFS	Crude Cells	Temperature-Treated Cells
**3 nm**	++	++	-	++	-	-
**1 nm**	++	++	-	++	-	-
**RSOLV**	++	++	-	++	-	-
**S4**	++	++	-	++	-	-
**pepp1**	++	++	-	++	-	-
**pepp2**	++	++	-	++	-	-

**Table 5 microorganisms-11-02679-t005:** Antibacterial activity of whole cultures, CFS, and cells from the selected *Lpb. plantarum* strains. The CFS were used crudely or after neutralization with NaOH; *Lpb. plantarum* cells were tested before and after heat treatment. The indicator bacteria was *Listeria monocytogenes* CECT 4031. No (-) or strong (++) inhibition showed a zone lower than 7 mm, ranging from 7 to 10 mm, or more than 10 mm, respectively. Assays were performed in duplicate.

	*L. monocytogenes*					
Strain	LAB Culture	Crude CFS	pH-Neutralized CFS	Temperature-Treated CFS	Crude Cells	Temperature-Treated Cells
**3 nm**	++	++	-	++	-	-
**1 nm**	++	++	-	++	-	-
**RSOLV**	++	++	-	++	-	-
**S4**	++	++	-	++	-	-
**pepp1**	++	++	-	++	-	-
**pepp2**	++	++	-	++	-	-

**Table 6 microorganisms-11-02679-t006:** Concentration of the main organic acids in CFS from 24 h cultures of the selected *Lpb. plantarum* strains grown in MRS broth, as determined using HPLC. Results (mg L^−1^) are presented as the mean and SD of the two measures.

	Tartaric Acid	Malic Acid	Ascorbic Acid	Lactic Acid	Acetic Acid	Succinic Acid	Fumaric Acid	**Citric Acid**
**MRS**	503.27 ± 20.85	3786.64 ± 95.81	51.80 ± 1.30	0.00 ± 0.00	17,906.53 ± 162.88	1066.96 ± 27.55	0.00 ± 0.00	1206.78
**3 nm**	1202.55 ± 30.22	1326.48 ± 176.26	100.09 ± 42.11	14,931.50 ± 133.44	16,421.03 ± 2 236.89	5758.93 ± 729.88	15.47 ± 6.71	2662.19 ± 403.92
**1 nm**	1347.92 ± 25.81	1808.24 ± 42.34	102.69 ± 5.66	14,620 ± 214.70	17,033.25 ± 877.62	11,693.70 ± 708.58	0.00 ± 0.00	1833.64 ± 134.04
**S4**	1218.73 ± 42.21	1294.83 ± 58.28	117.81 ± 1.86	15,419.91 ± 89.09	16,312.59 ± 1 242.27	9009.26 ± 1 374.46	0.00 ± 0.00	2224.48 ± 19.15
**RSOLV**	677.69 ± 128.32	1735.10 ± 171.48	0.00 ± 0.00	14,985.98 ± 195.53	19,853.59 ± 541.76	10,193.35 ± 562.11	0.00 ± 0.00	1050.29 ± 200.52
**pepp1**	1201.36 ± 89.76	1309.51 ± 9.18	135.44 ± 4.85	13,829.46 ± 132.96	19,888.39 ± 809.60	6897.52 ± 781.84	0.00 ± 0.00	446.71 ± 45.63
**pepp2**	1092.72 ± 116.94	1335.78 ± 99.67	0.00 ± 0.00	15,439.67 ± 140.01	42,409.27 ± 4 762.60	7336.16 ± 473.52	0.00 ± 0.00	894.15 ± 25.05

**Table 7 microorganisms-11-02679-t007:** Antifungal activity of *Lpb. plantarum* strains against *P. expansum, A. niger, F. culmorum* CECT 2148, *S. cerevisiae*, and *B. cinerea* CECT 20973, as determined using the overlay method. No (-), mild (+), or strong (++) inhibition showed a zone lower than 1 mm, ranging from 1 to 5 mm, or more than 5 mm, respectively. Assays were performed in duplicate.

Strain	*P. expansum*	*A. niger*	*B. cinerea* CECT 20973	*F. culmorum*	*S. cerevisiae*
**3 nm**	+	++	+	++	-
**1 nm**	+	++	+	+	-
**RSOLV**	++	+	++	++	-
**S4**	++	++	+	+	-
**pepp1**	++	++	+	+	-
**pepp2**	++	++	++	++	-

**Table 8 microorganisms-11-02679-t008:** Auto-aggregation ability of *Lpb. plantarum* strains after 4 h and 24 h incubation in 37 °C in PBS at pH 7. The assay was performed in duplicate.

	T4	T24
**3 nm**	24.31% ± 6.84%	53.46% ± 2.01%
**1 nm**	16.82% ± 2.33%	53.59% ± 12.61%
**S4**	18.19% ± 2.54%	70.30% ± 4.73%
**RSLOV**	28.87% ± 3.68%	60.97% ± 5.67%
**pepp1**	21.50% ± 1.75%	34.67% ± 0.99%
**pepp2**	16.70% ± 1.12%	69.82% ± 2.20%

**Table 9 microorganisms-11-02679-t009:** Co-aggregation of *Lpb. plantarum* index with the pathogen *Listeria monocytogenes* CECT 4031 and *E. coli* after 4 h and 24 h incubation at 37 °C in PBS pH7. Assays were performed in duplicate.

	*L. monocytogenes*		*E. coli*	
Strain	T4	T24	T4	T24
3 nm	7.94% ± 0.94%	7.28% ± 0.47%	17.41% ± 1.54%	43.42% ± 6.08%
1 nm	6.38% ± 1.21%	5.52% ± 0.18%	17.73% ± 7.75%	54.88% ± 6.98%
S4	7.47% ± 0.14%	7.37% ± 0.46%	15.05% ± 0.89%	36.96% ± 5.72%
RSOLV	7.169% ± 2.25%	8.76% ± 1.16%	18.27% ± 4.32%	51.15% ± 4.69%
pepp1	5.38% ± 0.00%	5.38% ± 1.34%	20.72% ± 2.67%	43.53% ± 0.00%
pepp2	4.99% ± 0.55%	4.61% ± 0.79%	28.30% ± 10.70%	43.16% ± 1.65%

**Table 10 microorganisms-11-02679-t010:** Antibiotic resistance assay. The experiment was performed in duplicate. The inhibition zone diameters were measured. Susceptibility was expressed as resistant (R), susceptible (S) and intermediate (I), as mentioned.

ATB/Strain	3 nm	1 nm	S4	RSOLV	pepp1	pepp2
**Ampicillin**	I	S	I	I	S	I
**Vancomycin**	R	R	S	R	R	R
**Gentamycin**	I	I	R	S	S	I
**Kanamycin**	R	R	R	R	R	R
**Streptomycin**	R	S	R	R	R	R
**Tetracyclin**	R	R	R	R	R	R
**Erythromycin**	I	R	R	I	R	I
**Clindamycin**	I	R	R	I	I	I
**Claromycin**	I	R	R	R	S	I

**Table 11 microorganisms-11-02679-t011:** Half-maximal inhibitory concentration (IC_50_) of different CFSs on the proliferation of LS cell line after 24 h and 72 h at 37 °C.

Strain	IC_50_ (after 24 h) mg/mL	IC_50_ (after 72 h) mg/mL
**3 nm**	11.58	5.06
**1 nm**	7.48	2.74
**S4**	12.06	4.96
**RSOLV**	10.80	2.25
**pepp1**	9.12	2.82
**pepp2**	11.67	2.18

**Table 12 microorganisms-11-02679-t012:** Pathogen antagonism by *Lpb. plantarum* on tomatoes. CFU counts of pathogens inoculated alone (control) or in the presence of each of the indicated *Lpb. plantarum* strains are reported. Mean ± standard deviation calculated from three different experiments. Each experiment was performed in duplicate. Two-way ANOVA and Dunnett’s multiple comparisons test were used to compare strains at different times. ^$^ *p* < 0.05 vs. 5 days of 1 nm–*Listeria* and 5 days of pepp1–*Listeria*; ^&^ *p* < 0.05 vs. 1 day of RSOLV–*E. coli* and 1 day of pepp1–*E. coli*.

	Number of Tomato-Attached Bacteria (Log CFU/g Tomato)			
	Medium Agar	1 Day	3 Days	5 Days
*Listeria* (control)	PALCAM	1.68 ± 1.39	1.67 ± 1.10	1.74 ± 0.70 ^$^
3 nm–*Listeria*	PALCAM	1.73 ± 0.87	1.44 ± 0.74	1.52 ± 1.48
1 nm–*Listeria*	PALCAM	1.70 ± 1.23	1.02 ± 0.62	0.00
S4–*Listeria*	PALCAM	1.46 ± 0.22	1.00 ± 0.67	0.00
RSOLV–*Listeria*	PALCAM	1.77 ± 0.80	1.614 ± 1.38	1.18
pepp1–*Listeria*	PALCAM	1.49 ± 0.997	1.54 ± 1.41	1.15 ± 0.45
pepp2–*Listeria*	PALCAM	1.70 ± 1.14	1.16 ± 0.84	1.63 ± 1.16
*E. coli* (control)	SMAC	1.92 ± 1.61 ^&^	1.72 ± 0.99	1.63 ± 1.54
3 nm–*E. coli*	SMAC	1.82 ± 1.13	1.55 ± 1.43	1.34 ± 0.94
1 nm–*E. coli*	SMAC	1.87 ± 1.24	1.70 ± 0.97	1.38 ± 0.31
S4–*E. coli*	SMAC	1.88 ± 1.05	1.73 ± 1.15	1.13 ± 0.45
RSOLV–*E. coli*	SMAC	1.87 ± 1.04	1.51 ± 1.33	1.56 ± 1.24
pepper1–*E. coli*	SMAC	1.81 ± 1.22	1.45 ± 1.11	1.24 ± 0.45
pepper2–*E. coli*	SMAC	1.84 ± 1.15	1.53 ± 0.96	1.22 ± 0.41

## Data Availability

The datasets generated for this study are available on request to the corresponding author.

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
