# Peer review of "Lactiplantibacillus plantarum from Unexplored Tunisian Ecological Niches: Antimicrobial Potential, Probiotic and Food Applications"

_microorganisms, 2023, doi:10.3390/microorganisms11112679_

Round 1
Reviewer 1 Report
Comments and Suggestions for Authors
The study is well designed and interesting. A minor revision is needed. The following suggestions may improve the manuscript:
L107 Therefore, bio-antimicrobial agents, such as antagonistic bacteria, have emerged as alternatives recently.
State which strains of antagonistic bacteria are the most potent in preventing contamination and spoilage of vegetables.
Many available studies aimed to isolate and identify epiphytic lactic acid bacteria (LAB) from fresh fruits and leafy vegetables and characterize their antagonistic capacity due to their ability to produce bacteriocins or antibacterial compounds. The authors should consider highlighting in the introduction which bacteria are most often used so far with the purpose of reducing/preventing contamination and spoilage of vegetables (for example Pediococcus pentosaceus and Latilactobacillus graminis).
L 127 24 h bacterial cultures from the isolated microbes were poured into each well.
Was the filtered supernatant of the overnight bacterial culture used? Or some kind of bacterial extract? It is necessary to describe this part in more detail already here (or simply label it as cell-free supernatant from bacterial cultures).
L262 human colorectal adenocarcinoma LS cells
LS 174T?
L351 analysis recommended by Prism.
analysis recommended by the software GraphPad Prism 6.0 (GraphPad Software Inc., San Diego, CA, USA).
L533 The biofilm formation was also investigated using Crystal Violet (Figure 4B).
…using Crystal Violet assay (Figure 4B).
L559 the bacterial medium (MRS) did not show any cytotoxic effect.
the bacterial medium (MRS) was used as a solvent (negative) control and did not show any cytotoxic effect.
L574 Half-maximal inhibitory concentration (IC50)
Half-maximal inhibitory concentration (IC50)
L754 Besides, in agreement with our findings, several authors demonstrated the ability of Lpb. plantarum to survive to OGI transit with a rate of 103 CFU·mL−1
Throughout the manuscript, viability results are presented as Log CFU/g. Please write Log values also in the discussion.
General comments:
-How did you choose the control strains? Why are there different control strains in different assays?
Author Response
Dear Editor,
we would like to thank the reviewers for their appreciation and their valuable comments. We have revised the ms according to their indications.
Below we respond point by point to the reviewers’ comments.
Reviewer 1
The study is well designed and interesting. A minor revision is needed. The following suggestions may improve the manuscript:
*L107 Therefore, bio-antimicrobial agents, such as antagonistic bacteria, have emerged as alternatives recently.
State which strains of antagonistic bacteria are the most potent in preventing contamination and spoilage of vegetables.
Many available studies aimed to isolate and identify epiphytic lactic acid bacteria (LAB) from fresh fruits and leafy vegetables and characterize their antagonistic capacity due to their ability to produce bacteriocins or antibacterial compounds. The authors should consider highlighting in the introduction which bacteria are most often used so far with the purpose of reducing/preventing contamination and spoilage of vegetables (for example Pediococcus pentosaceus and Latilactobacillus graminis).
R: thanks, we have integrated the ms with this indication and the following ref. were added: Iosca et al., Anti-Spoilage Activity and Exopolysaccharides Production by Selected Lactic Acid Bacteria. Foods, 2022.; Yin et al., Pre-harvest biocontrol of Listeria and Escherichia coli O157 on lettuce and spinach by lactic acid bacteria. Int J Food Microbiol, 2023; González-Pérez et al., Potential control of foodborne pathogenic bacteria by Pediococcus pentosaceus and Lactobacillus graminis isolated from fresh vegetables. Microbiology and Biotechnology Letters, 2019.
*L127 24 h bacterial cultures from the isolated microbes were poured into each well.
Was the filtered supernatant of the overnight bacterial culture used? Or some kind of bacterial extract? It is necessary to describe this part in more detail already here (or simply label it as cell-free supernatant from bacterial cultures).
*L262 human colorectal adenocarcinoma LS cells
LS 174T?
R: thanks, the cell line code has been added.
*L351 analysis recommended by Prism.
analysis recommended by the software GraphPad Prism 6.0 (GraphPad Software Inc., San Diego, CA, USA).
R: thanks, this has been added
*L533 The biofilm formation was also investigated using Crystal Violet (Figure 4B).
…using Crystal Violet assay (Figure 4B).
R: done.
*L559 the bacterial medium (MRS) did not show any cytotoxic effect.
the bacterial medium (MRS) was used as a solvent (negative) control and did not show any cytotoxic effect.
R: thanks, this has been corrected
*L574 Half-maximal inhibitory concentration (IC50)
Half-maximal inhibitory concentration (IC50)
R: thanks, corrected.
*L754 Besides, in agreement with our findings, several authors demonstrated the ability of Lpb. plantarum to survive to OGI transit with a rate of 103 CFU·mL−1
Throughout the manuscript, viability results are presented as Log CFU/g. Please write Log values also in the discussion.
R: thanks, done.
General comments:
-How did you choose the control strains? Why are there different control strains in different assays?
Reviewer 2 Report
Comments and Suggestions for Authors
In manuscript "Lactiplantibacillus plantarum from unexplored Tunisian ecological niches: antimicrobial potential, probiotic and food applications" a few other potentially probiotic bacteria have been described. What makes it interesting is the potential application of these potentially probiotic bacteria on the matrix food model and on the surface of the tomato.
I have a few questions:
I don't understand the procedure in line 242? Why did you use biofilm staining with resazurin?
I don't understand why aggregation and co-aggregation was done at 37°C?
And explain how the % of adhesion to Caco2 cells is determined?
Author Response
Dear, we would like to thanks for your suggestions/critical advices and your valuable comments as well. We have revised the ms according to your indications.Below we respond point by point to the reviewers’ comments.
Reviewer 2
In manuscript "Lactiplantibacillus plantarum from unexplored Tunisian ecological niches: antimicrobial potential, probiotic and food applications" a few other potentially probiotic bacteria have been described. What makes it interesting is the potential application of these potentially probiotic bacteria on the matrix food model and on the surface of the tomato.
I have a few questions:
*I don't understand the procedure in line 242? Why did you use biofilm staining with resazurin?
R: The biofilm production was quantified by two methods: the classic crystal violet method allowed us to stain both viable and non viable adherent bacterial, while by using the redox dye resazurin we aimed at detecting the metabolic activity of the microbial cells, therefore quantifying only the viable adherent cells. Resazurin has been often used to quantify viable microbial biofilm [Van den Driessche et al. Optimization of resazurin-based viability staining for quantification of microbial biofilms. J Microbiol Methods. 2014 Mar;98:31-4. doi: 10.1016/j.mimet.2013.12.011; Cruz et al. Defining conditions for biofilm inhibition and eradication assays for Gram-positive clinical reference strains. BMC Microbiol 18, 173 (2018). https://doi.org/10.1186/s12866-018-1321-6 ].
Thanks for this comment: we explained this better also in the main text of the revised version.
*I don't understand why aggregation and co-aggregation was done at 37°C?
R: According to previous literatures [Leska A et al., 2023 https://doi.org/10.3390/molecules27248945;Tuo Y et al., 2013 https://doi.org/10.3168/jds.2013-6547], aggregation ability is related to adhesion ability to intestinal epithelial cells. We selected 37°C because it is the ideal/optimal temperature for bacterial growth (LAB, E.coli and Listeria) and it is the temperature we used for the adhesion test on caco2 cells. In order to have a logic comparison to published results, the standard protocol works at the temperature of bacterial growth.
*And explain how the % of adhesion to Caco2 cells is determined?
Thanks, this has been better explained in the material and method section (par 2.9) and in legend to figure 6 of the revised manuscript.